# One-Pot Synthesis of Phosphinylphosphonate Derivatives and Their Anti-Tumor Evaluations

**DOI:** 10.3390/molecules26247609

**Published:** 2021-12-15

**Authors:** Jade Dussart-Gautheret, Julia Deschamp, Thibaut Legigan, Maelle Monteil, Evelyne Migianu-Griffoni, Marc Lecouvey

**Affiliations:** Department of Chemistry, Université Sorbonne Paris Nord, CSPBAT, CNRS, UMR 7244, 1 rue de Chablis, F-93000 Bobigny, France; jade92.dussart@gmail.com (J.D.-G.); julia.deschamp@univ-paris13.fr (J.D.); thibaut.legigan@univ-paris13.fr (T.L.); maelle.monteil@univ-paris13.fr (M.M.)

**Keywords:** phosphinylphosphonate, bisphosphonate, bisphosphinate, synthesis, anti-tumor activity

## Abstract

This paper reports on the synthesis of new hydroxymethylene-(phosphinyl)phosphonates (HMPPs). A methodology has been developed to propose an optimized one-pot procedure without any intermediate purifications. Various aliphatic and (hetero)aromatic HMPPs were synthesized in good to excellent yields (53–98%) and the influence of electron withdrawing/donating group substitution on aromatic substrates was studied. In addition, the one-pot synthesis of HMPP was monitored by ^31^P NMR spectroscopy, allowing effective control of the end of the reaction and identification of all phosphorylated intermediate species, which enabled us to propose a reaction mechanism. Optimized experimental conditions were applied to the preparation of biological relevant aminoalkyl-HMPPs. A preliminary study of the complexation to hydroxyapatite (bone matrix) was carried out in order to verify its lower affinity towards bone compared to bisphosphonate molecules. Moreover, in vitro anti-tumor activity study revealed encouraging antiproliferative activities on three human cancer cell lines (breast, pancreas and lung).

## 1. Introduction

The synthesis of phosphorus-containing compounds still represents a major challenge in the development of more efficient therapeutics [1,2,3]. Among phosphorylated molecules, 1-hydroxymethylene-1,1-bisphosphonates (HMBPs **II**) are C-substituted pyrophosphate **I** analogues used to treat and prevent osteoporosis, cancer and bone metastasis (Figure 1) [4,5,6]. In addition, these molecules have shown interesting antitumor properties on in vitro and in vivo models of soft tissue primary tumor [7,8,9]. However, the use of HMBPs **II** is limited due to their poor oral bioavailability and biodistribution [10]. Indeed, HMBPs **II** are very polar molecules because of their two or three negative charges at physiological pH which imply that they have limitations in terms of their cell uptake [11]. The synthesis of more lipophilic analogues such as hydroxy-methylenebis(*H*-phosphinates) (HMBPi **III**) or hydroxymethylene(phosphinyl)phosphonates (HMPPs **IV**) should overcome these drawbacks (Figure 1).

The afore-mentioned molecules will present fewer negative charges, which should imply a lower hydrophilicity and acidity (similar to carboxylic acid). In this context, we have recently reported the efficient synthesis of substituted HMBPi **III** by developing a convenient one-pot methodology [12,13,14].

In the meantime, the synthesis of hydroxy-methylene(phosphinyl)phosphonates HMPPs **IV** has not been reported yet. As far as we know, only a few studies have prospectively documented the access to the methylene(phosphinyl)phosphonate core by multi-step syntheses [15,16,17,18,19,20]. These strategies mostly consisted of firstly deprotonating a dialkyl methylphosphonate derivative to form the corresponding carbanion. Thereafter, they involved adding an alkylphosphonochloridate to yield the corresponding methylene(phosphinyl)phosphonate (Figure 1) [15,16,17,18].

Inspired by our previous works on HMBPs **II** or HMBPi **III**, we focused on the synthesis of the hydroxymethylene(phosphinyl)phosphonate scaffold. HMBPs **II** [21] or HMBPi **III** [12,13,14] were actually synthesized by the successive double addition of tris(trimethylsilyl)-phosphite (TTSP, R^2^ = OTMS) or bis(trimethylsilyl)phosphonite (BTSP, R^2^ = H) onto an acyl chloride, via a silylated α-ketophosphonate or α-ketophosphinate intermediate, respectively (Figure 1). Our novel strategy consists of obtaining hydroxymethylene(phosphinyl)phosphonate derivatives **IV** in three steps via the similar in situ formation of a silylated α-ketophosphonate (Figure 1) [21,22].

From a strategic point of view, the reaction between an acyl chloride and TTSP could theoretically be considered to firstly furnish a silylated α-ketophosphonate intermediate which could subsequently react with BTSP to yield the corresponding HMPP **IV** after methanolysis (Figure 1). However, the direct addition of only one equivalent of TTSP onto an acyl chloride cannot lead properly to the sole silylated α−ketophosphonate. Indeed, this intermediate is highly reactive and can react itself with TTSP to form the corresponding HMBP **II**. This overreaction can be limited, but unfortunately, it takes place even at a lower temperature [22]. In our case, this side-reaction will lead, in the end, to an inseparable mixture of HMBP **II** and HMPP **IV**. To prevent the HMBP formation, firstly, the reaction between less reactive trimethylphosphite and an acyl chloride can be considered to prepare the dimethyl ester α-ketophosphonate exclusively. This diester is rather stable and can be easily isolated and stored for several months. Then, the dealkylation can be achieved by using a silylating agent to generate the corresponding silylated α-ketophosphonate, which can undergo the attack of BTSP to finally provide the expected HMPP **IV** disodium salt after methanolysis and basic treatment. Herein, we report our methodological development to efficiently access various substituted HMPPs **IV**.

## 2. Results and Discussion

### 2.1. Chemistry

#### 2.1.1. Development of an Optimized One-Pot Sequence

First of all, the synthesis of the dimethyl ester α-ketophosphonate **3a** was performed by adding dropwise acetyl chloride **2a** onto trimethylphosphite **1** at 0 °C (Figure 2: pathway (a)). The reaction was completed after 2.5 h at room temperature. The end of the reaction was monitored by ^31^P NMR spectroscopy by following the disappearance of **1** signal (141.1 ppm) in the favor of the **3a** signal emergence (0.6 ppm). The dimethyl α-ketophosphonate **3a** was then isolated in an excellent yield (98%) after distillation under reduced pressure.

Thereafter, the dealkylation step was carried out in the presence of a required excess of silylating agent (Figure 2: pathway (b)). We observed that bromotrimethylsilane (TMSBr) can effectively trigger the dealkylation of **3a** to generate the silylated compound **4a** exhibiting a shift in ^31^P NMR signal in negative values (−17.8 ppm) [23]. After evaporating the volatile fractions under reduced pressure, the silylated HMPP was rapidly formed by dropwise adding the silylated phosphonite **5** for 20 min at 0 °C. The reaction had already been completed since the end of the BTSP **5** addition. Then, a methanolysis step followed by a pH adjustment to 7 yielded **6a** as disodium salt in an excellent yield of 91% after easy purification by precipitation. For the latter step, the preparation of bis(trimethylsilyl)phosphonite **5** can be smoothly achieved by the silylation of hypophosphorous acid in the presence of *N,O*-bis(trimethylsilyl)acetamide (BSA) as previously reported [12,13,14,21,24].

Next, the same sequential process was conducted without proceeding the evaporation at the end of the dealkylation step. In this case, HMPP **6a** was obtained as well as a significant amount of the corresponding HMBP (**6a**/HMBP: 85/15). Unfortunately, the mixture remained inseparable despite the purification attempts. We postulated that the presence of HMBP resulted from the oxidation of HMPP, probably due to the excess of TMSBr.

Afterwards, we attempted to perform the dealkylation in the presence of BSA. However, the silylated phosphonate **4a** was not formed. Although BSA is often used to promote reactions of silylation, it cannot trigger the dealkylation.

The concomitant addition of bromotrimethylsilane and BTSP **5** onto the dimethyl ester α-ketophosphonate **3a** was also carried out at 0 °C (Figure 2: pathway (c)). In this case, the ^31^P NMR reaction monitoring had shown the disappearance of **3a** and **5** signals and numerous arising signals. After evaporation, methanolysis and basic treatment, a complex mixture of phosphonylated compounds was obtained including a minor quantity of HMPP **6a**.

We also tested the addition of BTSP **5** onto **3a** followed by the dealkylation (Figure 2: pathway (d)). The first step allowed us to successfully obtain the silylated dimethyl ester HMPP **7a** for 20 min at 0 °C. After that, the dealkylation partially occurred in the presence of bromotrimethylsilane (2.5 equivalents) with a conversion of 63% only. Unfortunately, the conversion was not enhanced in the presence of 5 or 10 equivalents of TMSBr. It was noted that the subsequent methanolysis of **7a** can lead to dimethyl ester HMPP. Finally, the synthesis of **6a** was conducted in a one-pot procedure starting from **1** and **2a** to form in situ **3a,** which was readily dealkylated by TMSBr. The resulting silylated compound **4a** subsequently underwent an attack of BTSP **5** to form the expected silylated HMPP (Figure 2, (e) versus (a,b)). The methanolysis and pH adjustment (pH = 7) obtained **6a** in an excellent yield (91%) after purification (Figure 2: pathway (e)). The latest adequate conditions enabled us to propose an easily handled one-pot process to provide HMPP derivatives which did not require any intermediate purifications.

#### 2.1.2. NMR Monitoring of the One-Pot Reaction

Figure 2 depicts the detailed ^31^P NMR monitoring of the total process that allowed us to propose the one-pot sequence. When acetyl chloride **2a** was added to trimethylphosphite **1**, its signal (141.1 ppm) disappeared in favor of an emerging new signal at −0.6 ppm corresponding to the dimethyl α-ketophosphonate **3a** (Figure 2, spectra (b) versus (a)). During the following dealkylation step in the presence of TMSBr, the signal of **3a** shifted in the negative chemical region (−17.8 ppm) which supported the formation of the silylated α-ketophosphonate **4a** (Figure 2, spectra (c) versus (b)). Meanwhile, the preparation of BTSP **5** resulting from the silylation of hypophosphorous acid by BSA was simultaneously monitored by ^31^P and {^1^H}^31^P NMR spectroscopy as the signal of H_3_PO_2_ at 12.8 ppm faded away in favor of the appearance of a new peak at 141.6 ppm in the trivalent phosphorus region (Figure 2, spectra (d)). The doublet with a large coupling constant featured in the ^31^P spectrum corroborated the presence of the P-H bond. Thereafter, the monitoring of the subsequent addition of **5** onto **4a** showed the fast disappearance of their signals and the formation of several peaks in the pentavalent phosphorus region (Figure 2, spectra (e) versus (c,d)). The two new sets of peaks at ~20 ppm and 1.5 ppm were assigned to **8a,** which possesses two non-equivalent phosphorus atoms. The presence of a parasite peak was detected at −13.9 ppm (<7%) fitting to (TMSO)_2_P(O)H resulting from the oxidation of **5**. After the final methanolysis and pH adjustment at 7, the two signal sets of **8a** moved to 27.3 and 17.8 ppm as two thin doublets (*^2^J_P-P_*= 30.0 Hz) corresponding to the crude HMPP **6a** (Figure 2, spectra (f) versus (e)). Regarding the different coupling constants in the ^31^P NMR spectrum, the two signals could be assigned, respectively, to the phosphinate moiety at 27.3 ppm (large *^1^J*= 527.8 Hz, P–H) and the phosphonate group at 17.8 ppm. Finally, the easy purification of the crude HMPP **6a** by precipitation enabled us to discard the disodium phosphonate by-product to furnish pure HMPP **6a** (Figure 2, spectra (g) versus (f)).

#### 2.1.3. Scope of the Reaction

We then studied the scope of the one-pot reaction in the presence of various acyl chlorides according to the previous optimal process (Figure 3).

The above optimized conditions were tested on other aliphatic acyl chlorides **2b** (R = *i-*Bu) and **2c** (R = *n*-pentyl). In both cases, the corresponding HMPP **6b** and **6c** were isolated in excellent yields after purification. The reaction has also been transposed on aromatic substrates. When the reaction was carried out on benzoyl chloride **2d** (R = Ph), the corresponding HMPP **6d** was obtained in an excellent yield (95%).

We then evaluated the reactivity of *para*-substituted aromatic acyl chlorides **2e–h**. As a general trend, the substitution at the *para-* position by an electron withdrawing or an electron donating group had little influence on the course of the reaction. In these cases, the expected HMPP **6e–h** were synthesized in good to excellent yields (82–92%).

The reaction was also conducted on *ortho*-substituted acyl chlorides **2i** (*o*-Me), **2j** (*o*-OMe), **2k** (*o*-F) and 1-naphthoyl chloride **2l**. The first attempt under the previous optimal conditions unfortunately led to a mixture of the expected HMPP **6i–l** and a phosphono-phosphonate side product **9i–l** (**6i**/**9i**: 40/60; **6j**/**9j**: 43/57; **6k**/**9k**: 50/50; **6l**/**9l**: 20/80). In our previous studies on HMBPs and HMBPi, the formation of this by-product resulting from a rearrangement had already been observed and it particularly occurred starting from bulkier or/and electron withdrawing acyl chlorides [12,21,25,26]. Several investigations indicated that the transposition could take place in the presence of a large quantity of the silylated α-keto-phosphonylated intermediates. Therefore, the reactions were performed by adding dropwise the silylated α-ketophosphonate **4i–l** onto the in situ BTSP **5** in order to disfavor the transposition. Additionally, this time, we were pleased that HMPP **6i–l** were predominantly produced (**6i**/**9i**: 80/20; **6j**/**9j**: 70/30; **6k**/**9k**: 70/30; **6l**/**9l**: 57/43). After purification, HMPP **6i**, **6j** and **6l** were isolated in rather good yields of 75%, 65% and 53%, respectively. Unfortunately, the *ortho*-fluoro-HMPP **6k** remained inseparable from the crude mixture despite many efforts.

The optimized conditions were also applied on heteroaromatic 2-thenoyl chloride **2m** and we were satisfied to observe the formation of the corresponding HMPP **6m** in a very good yield (82%).

Finally, the introduction of a benzyl group on the HMPP scaffold was considered as well. However, our standard conditions could not be used regarding our previous works on the synthesis of monomethyl ester HMBP [22,27]. Indeed, the reaction between phenylacetyl chloride **2n** and trimethylphosphite **1** led to an inseparable mixture of the expected dimethyl ester α-ketophosphonate **3n** and its corresponding enolic form which were obtained, respectively, as minor and major products regardless of the various conditions. However, this drawback was overcome by performing the reaction in the presence of TTSP at −70 °C. We decided to try the same strategy for the synthesis of HMPP **6n** despite the fact that the corresponding HMBP should also be formed as a by-product. Here, TTSP (1 equivalent) was dropwise added to phenylacetyl chloride **2n** at −70 °C which successively promoted the formation of **4n**. The reaction was carefully monitored by ^31^P NMR spectroscopy in order to avoid the second attack of TTSP on the corresponding silylated α-ketophosphonate **4n** as previously mentioned. The spectra showed fast signal disappearance related to TTSP (114.0 ppm) and an emerging new signal (−18.0 ppm) corresponding to **4n** in only 30 min without any trace of the enolic form or HMBP. The subsequent addition of BTSP **5** at −70 °C provided the expected benzyl substituted silylated HMPP in only 20 min. After methanolysis, pH adjustment and purification, HMPP **6n** was finally isolated in an excellent yield of 90%.

#### 2.1.4. Postulated Mechanism

According to our previous works and the present observations, we can postulate the following mechanism depicted in Figure 4.

First of all, the Arbuzov reaction between trimethylphosphite **1** onto the acyl chloride **2** could give the corresponding dimethyl ester α-ketophosphonate **3** via a tetravalent phosphonium intermediate. Thereafter, the concerted dealkylation of **3** in the presence of bromotrimethylsilane could in situ generate the silylated α-ketophosphonate **4** which subsequently could undergo the attack of BTSP **5** to form **10**. The trans-silylation of **10** could finally afford the silylated HMPP **8** (Figure 4, pathway (a) in black). Then, the isolation of the expected HMPP **6** could be achieved after methanolysis and a pH adjustment. Moreover, we can also propose two side routes that could explain the formation of phosphono-phosphonate isomers **9i–l** (Figure 4, pathways (b) and (c) in red). Because of the steric hindrance of **4i–l**, we could first envision the direct nucleophilic attack of BTSP **5** on its carbonyl oxygen atom, which could give a stabilized benzylic carbanion **12** (pathway (b)) as previously reported in the literature [28,29].

However, we can also consider another plausible pathway (c) in which the silylated phosphonium **10** could also form a three-membered ring intermediate **11** which could rapidly decompose into the same carbanion **12 [30]**. In both pathways (b) and (c), the negative charge in benzylic position of **12** is particularly stabilized in the presence of electron withdrawing substituents, which could explain our observations. Finally, the phosphonium intermediate **12** could be transformed in a silylated pentavalent phosphono-phosphonate **13** which would furnish the isomers **9i–l** after methanolysis and pH adjustment.

#### 2.1.5. Synthesis of Aminoalkyl-HMPPs

Regarding our methodological development, we turned our attention toward the synthesis of biological relevant aminoalkyl-HMPPs alendrionate **14o** and neridrionate **14p**. The latter are HMBP alendronate and neridronate analogues, which are both clinically used in bone diseases and exhibited in vitro anti-tumor properties (Figure 5).

First of all, 4-azidobutanoyl and 6-azidohexanoyl chlorides **2o-p** were synthesized starting from the corresponding 4-azidobutanoic and 6-azidohexanoic acids [12]. Then, **2o-p** were mixed with trimethylphosphite **1** to produce the azidoalkyl-α-ketophosphonates **3o-p, respectively**. The in situ dealkylation of **3o-p** triggered by 2.5 equivalents of bromotrimethylsilane gave the corresponding silylated azidoalkyl-α-ketophosphonates **4o-p**. The subsequent addition of BTSP **5** finally allowed the silylated azidoalkyl-HMPPs to be formed, which were treated by methanol to furnish the desired azidoalkyl-HMPPs **6o-p**, respectively, in excellent yields of 83% and 85% after purification and pH adjustment (pH = 7). Afterwards, the HMPP disodium salts **6o-p** were acidified by using a commercially available Dowex^®^ H^+^ resin in order to avoid palladium complexation. The HMPP acidic derivatives then underwent a hydrogenolysis step in the presence of dihydrogen and palladium on charcoal. The end of reaction was monitored by ^31^P NMR spectroscopy eventually. The expected aminoalkyl-HMPPs **14o-p** were finally isolated in very good yields (87–84%) after pH adjustment at 7.

### 2.2. Complexation Study

We then wanted to evaluate the anti-tumor properties of these new aminoalkyl-HMPPs on in vitro soft tissue primary tumor cells. However, before biological evaluation, we ensured HMPPs **14** are less affine for bone than HMBPs themselves. Indeed, as previously mentioned, HMBPs are used to treat bone pathologies because they can form strong complexes with calcium cation widely present in bone hydroxyapatite (HA) [5,6,31]. Despite this affinity, they have also shown promising anti-tumor activities towards vascularized tumors. However, their use is limited in this context due to their poor biodistribution and HMPPs could be a promising alternative to this drawback. We firstly studied the complexing properties of the new aminoalkyl-HMPP **14o** and the HMBPi analogue on HA in comparison with HMBP alendronate.

A solution of the phosphorylated compound in PBS (phosphate-buffered saline) was mixed with HA at 37 °C for 3 h. At t = 15, 70 and 180 min, the mixture was centrifuged and a ^31^P NMR analysis of the supernatant was carried out. The percentages of free HMBP, HMBPi or HMPP in solution were then determined (Figure 3).

From 15 min, the HA complexation ratios were 61% and 54% for alendronate and alendrionate, respectively, versus only 14% for alendrinate. Complexing ratios then quickly reached a plateau around 50–70 min at 71%, 61% and 23% for HMBP, HMPP and HMBPi analogues, respectively. The alendronate results were in accordance with published data on esterified HMBP affinity for HA [32]. As expected, we verified that HMPP and HMBPi had slower fixation and lower affinity for synthetic calcium phosphate compared to HMBP. It was also interesting to point out that HA complexation rate of alendrionate **14o** after 70 min was the same for alendronate after only 15 min.

### 2.3. Antiproliferative Evaluation

The in vitro effects of aminoalkyl-HMPP analogues alendrionate **14o** and neridrionate **14p** on human cancer cell growth were finally investigated for three different cell lines: MDA-MB-231 (breast), MIA PaCa-2 (pancreas) and A549 (lung). The MDA-MB-231 cell line is a highly aggressive form of human triple-negative breast cancer with limited treatment options. On the other hand, MIA PaCa-2 and A549 cell lines are relevant models for studying pancreas and lung cancers, respectively, which are characterized by low 5-year survival rates and limited therapeutic solutions. HMPP analogue activities were compared with those of alendronate and alendrinate for **14o** and neridronate and neridrinate for **14p**. Zoledronate was chosen as a reference [4]. Cancer cells were treated with the different compounds at various concentrations for 72 h incubation. Then, cell viability was measured by MTT assay and the corresponding IC_50_ values (concentration of the compound where the response is reduced by half) were determined (Table 1). Zoledronate showed lower IC_50_ values on three cancer cell lines in accordance with those previously reported in the literature [33].

As a general trend, the best antiproliferative effects were obtained on the A549 cell line with IC_50_ < 100 µM for all analogues except neridrinate. All compounds were inactive (IC_50_ > 100 µM) on the MIA PaCa-2 cell line except alendronate, showing rather good efficiency (IC_50_ = 17.2 µM). This HMBP displayed a similar activity on MDA-MB-231 cells (IC_50_ = 20.6 µM) while only alendrinate and neridrionate **14p** displayed IC_50_ values < 100 µM on this cell line. Alendronate and its analogues were more potent for inhibiting cancer cell growth than neridronate and derivatives (excepted neridrionate **14p** on A549 cells). These observations are in agreement with those already reported [34].

In the “alen” series, alendronate showed the best antiproliferative action on the three tested cell lines while alendrinate and alendrionate **14o** gave quite similar activities.

In the “neri” series, neridrinate was inactive on the three tested cell lines, whereas neridrionate **14p** exhibited encouraging antiproliferative activities on MDA-MB-231 and A549 cells, IC_50_ = 72 and 26.6 µM, respectively. The non-expected inhibitory activity of neridrionate **14p,** contrary to inactive neridronate, may suggest that long carbon chain HMPP could act on another site, or even another enzymatic target, rather than HMBP and HMBPi. Further enzymatic and docking studies would be required to investigate this intriguing result. Finally, the change in one of the two phosphonate moieties seems to strongly decrease the antiproliferative activity, indicating that the conservation of a phosphonate function is essential. This decrease in biological activity would be compensated in vivo by a better bioavailability, as suggested by the HA complexing study.

## 3. Materials and Methods

### 3.1. General Information

Reagents were purchased from common commercial suppliers (*Sigma-Aldrich, Alfa Aesar, Acros Organics*) and used as delivered. All solvents were extra-dried grade prior to use. *N,O*-*bis*(trimethylsilyl)acetamide (BSA) was purchased from *Alfa Aesar* (LOT: J24T014). Anhydrous H_3_PO_2_ was prepared from commercially aqueous H_3_PO_2_ solution (50% *w*/*w*) according to the procedure reported by Montchamp et al. [35]. Reactions requiring inert conditions were carried out in flame-dried glassware under an argon atmosphere. Specifically, **3a** was synthesized according the previously reported procedure [22]. Bis(trimethylsilyl)phosphonite **5**, alendrinate and neridrinate were synthesized according our previous reported procedure [12]. Alendronate, neridronate and zoledronate were obtained according the reported procedure [36].

NMR spectra were recorded at 20 °C on a Bruker Avance-III-400 spectrometer (^1^H: 400 MHz, ^13^C: 101 MHz, ^31^P: 162 MHz, ^19^F: 377 MHz). Chemical shifts (δ) were given in ppm, the number of protons (*n*) for a given resonance was indicated by nH, and coupling constants *J* in Hz. ^1^H NMR spectra were calibrated on non-deuterated solvent residual peak (H_2_O: 4.79 ppm), while H_3_PO_4_ (85% in water) was used as an external standard for ^31^P NMR. The following abbreviations were used for ^1^H, ^13^C, ^31^P and ^19^F NMR spectra to indicate the signal multiplicity: s (singlet), bs (broad singulet), d (doublet), bd (broad doublet), t (triplet), dd (doublet of doublets), dm (doublet of multiplet), m (multiplet), dq (doublet of quartets) and ddq (doublet of doublets of quartets). All ^13^C NMR spectra were measured with ^1^H-decoupling while ^31^P and ^19^F NMR spectra were measured with ^1^H coupling and ^1^H decoupling. ^1^H experiments with water presaturation were performed with D_1_ = 2 s and 128 scans. The reactions were followed by ^31^P and ^31^P{^1^H} NMR experiments (the spectra were recorded without lock and shims). All NMR peak assignments were performed thanks to 2D NMR COSY, HMQC and HMBC experiments. High-resolution mass spectra (HRMS) were performed on a Bruker maXis mass spectrometer in negative (ESI-) mode (ESI) by the “Fédération de Recherche” ICOA/CBM (FR2708) platform. MS analyses were performed using a Q-TOF Impact HD mass spectrometer equipped with the electrospray (ESI) ion source (Bruker Daltonics). The instrument was operated in the negative mode with an ESI source on a Q-TOF mass spectrometer with an accuracy tolerance of 2 ppm. Samples were diluted with acetonitrile and water (15:85) and were analyzed by mass spectrometry in continuous infusion using a syringe pump at 200 µL/min. The mass profiles obtained by ESI-MS were analyzed using DataAnalysis software (Bruker Daltonics).

### 3.2. Chemistry

#### 3.2.1. General Procedure Pathway (b) with Evaporation

Trimethylphosphite **1** (15.3 mL, 0.130 mol, 1.00 equiv.) was introduced to a dry and argon flushed 100 mL three-necked flask equipped with a thermometer and an argon inlet. Acetyl chloride **2a** (0.130 mol, 1.00 equiv.) was added dropwise at 0 °C under argon atmosphere and the mixture was stirred at room temperature for 2h. The reaction conversion was monitored by ^31^P NMR spectroscopy. The crude product **3a** was purified by distillation under pressure to give a pure oil (b.p.: 33–35 °C, 0.1 Torr).

Bromotrimethylsilane (1.65 mL, 12.5 mmol, 2.50 equiv.) was added dropwise to a solution of α-ketophosphonate dimethyl ester **3a** (5.00 mmol, 1.00 equiv.) in THF (1.00 mL) at 0 °C under argon. The reaction mixture was stirred at room temperature for 2 h. The reaction conversion was monitored by ^31^P NMR spectroscopy. The evaporation of volatile fractions followed by 3 co-evaporations with diethyl ether, gave bis(silylated) α-ketophosphonate **4a**. The silylated phosphonite **5** (5.00 mmol, 1.00 equiv.) was then added dropwise to the **4a** at 0 °C under argon. The reaction conversion was already completed after the addition of **5**. The volatile fractions were evaporated (with 3 co-evaporations with diethyl ether). The reaction mixture was quenched with methanol (10 mL) and the mixture was stirred for 30 min. Then, methanol was evaporated under reduced pressure. The residue was dissolved in a minimum of water and the pH was adjusted to 7.0 with an aqueous solution of sodium hydroxide (1 M). The solution was lyophilized. The crude residue was dissolved in a minimum of water then methanol was slowly added (30.0 mL) to form a precipitate which was recovered by filtration and dried in a vacuum dryer to give a pure solid **6**.

#### 3.2.2. General Procedure for Pathway (e)

Trimethylphosphite **1** (0.590 mL, 5.00 mmol, 1.00 equiv.) was introduced to a dry and argon flushed 100 mL three-necked flask equipped with a thermometer and an argon inlet. The adequate acyl chloride **2** (5.00 mmol, 1.00 equiv.) was added dropwise at 0 °C under argon atmosphere and the mixture was stirred at room temperature for 2 h–2.5 h. The reaction conversion was monitored by ^31^P NMR spectroscopy. Then, bromotrimethylsilane (1.65 mL, 12.5 mmol, 2.50 equiv.) was directly added to a solution of **3** in THF (1.00 mL) at 0 °C under argon. The reaction mixture was stirred at room temperature for 2 h. The reaction conversion was monitored by ^31^P NMR spectroscopy and the evaporation of volatile fractions followed by 3 co-evaporations with diethyl ether gave bis(silylated) α-ketophosphonate **4**. The silylated phosphonite **5** (5.00 mmol, 1.00 equiv.) was then added dropwise to **4** at 0 °C under argon. The reaction conversion was already completed after addition of **5**. The reaction mixture was quenched with methanol (10.0 mL) and the mixture was stirred for 30 min. Then, solvents were evaporated under reduced pressure. The residue was dissolved in a minimum of water and the pH was adjusted to 7.0 with an aqueous solution of sodium hydroxide (1 M). The solution was lyophilized. The crude residue was dissolved by a minimum of water followed by addition of methanol (30 mL). The formed precipitate is recovered by filtration and dried to give a pure solid **6**.

#### 3.2.3. General Synthesis Procedure for **6i–j,l**

Trimethylphosphite **1** (0.590 mL, 5.00 mmol, 1.00 equiv.) was introduced to a dry and argon flushed 100 mL three-necked flask equipped with a thermometer and an argon inlet. The adequate acid chloride **2** (5.00 mmol, 1.00 equiv.) was added dropwise at 0 °C under argon atmosphere and the mixture was stirred at room temperature for 2h-2h30. The reaction conversion was monitored by ^31^P NMR spectroscopy. Then, bromotrimethylsilane (1.65 mL, 12.5 mmol, 2.50 equiv.) was directly added to a solution of **3** in THF (1.00 mL) at 0 °C under argon. The reaction mixture was stirred at room temperature for 2 h. The reaction conversion was monitored by ^31^P NMR spectroscopy and the evaporation of volatile fractions followed by 3 co-evaporations with diethyl ether gave bis(silylated) α-ketophosphonate **4**. The bis(silylated) α-ketophosphonate **4** was then added dropwise at 0 °C under argon to another dry and argon flushed 100 mL three-necked flask equipped with a thermometer and an argon inlet, and containing the silylated phosphonite **5** (5.00 mmol, 1.00 equiv.). The reaction conversion was already completed after the addition of **4**. The reaction mixture was quenched with methanol (10.0 mL) and the mixture was stirred for 30 min. Then, the solvents were evaporated under reduced pressure. The residue was dissolved in a minimum of water and the pH was adjusted to 7.0 with an aqueous solution of sodium hydroxide (1 M). The solution was lyophilized. The crude residue was dissolved in a minimum of water, then methanol was slowly added (30.0 mL) to form a precipitate, which was recovered by filtration and dried to give a pure solid **6i–j,l**.

#### 3.2.4. General Synthesis Procedure for **6n**

tris(trimethylsilyl)phosphite (1.67 mL, 5.00 mmol, 1.00 equiv.) **1a** was added dropwise to a cooled solution of acid chloride **2n** (5.00 mmol, 1.00 equiv.) in THF (10.0 mL) at −70 °C under argon. The reaction mixture was stirred at −70 °C for 30 min and silylated phosphonite **5** (5.00 mmol, 1.00 equiv.) was then added dropwise at the same temperature. The reaction was already completed after the addition of **5**. The reaction mixture was quenched with methanol (10.0 mL) and the mixture was stirred for 30 min. Then, the solvents were evaporated under reduced pressure. The residue was dissolved in a minimum of water and the pH was adjusted to 7.0 with an aqueous solution of sodium hydroxide (1 M) and the solution was lyophilized. The crude residue was dissolved in a minimum of water, then methanol was slowly added (30.0 mL) to form a precipitate, which was recovered by filtration and dried to give a pure solid **6n**.

**1-Hydroxyethane-1,1-(H-phosphinylphosphonate) disodium salt 6a.** White powder (1.06 g, 91% yield (from pathway (b) with evaporation), 1.06 g, 91% yield (from pathway (e)); IR (neat, cm^−1^) ν = 3415 br, 2955 w, 2884 w, 2853 w, 1150 m, 1040 s, 848 s, 733 m ^31^P{^1^H} NMR (162 MHz, D_2_O) δ 27.3 (d, *^2^J_P-P_* = 31.7 Hz), 17.2 (d, *^2^J_P-P_* = 31.7 Hz); ^31^P NMR (162 MHz, D_2_O) δ 27.3 (ddq, *^1^J_P-H_* = 526.8 Hz, *^2^J_P-P_* = 31.7 Hz, *^3^J_P-H_* = 14.8 Hz), 17.2 (dq, *^2^J_P-P_* = 31.7 Hz, *^3^J_P-H_* = 14.8 Hz); ^1^H NMR (400 MHz, D_2_O) δ 6.83 (d, *^1^J_P-H_* = 526.8 Hz, 1H, P-H), 1.36 (t, *^3^J_P-H_* = 14.3 Hz, 3H); ^13^C NMR (101 MHz, D_2_O) δ 71.8 (dd, *^1^J_P-C_* = 140.3 Hz, 94.3 Hz), 17.2; MS (ESI^−^) m/z: 188.97 [M-H]^−^, 210.95 [M-2H+Na]^−^, 170.96 [M-H-H_2_O]^−^, 122.99 [M-H-H_3_PO_2_^−^; HRMS (ESI^−^) *m*/*z*: [M-H]^−^ Calcd. for C_2_H_7_O_6_P_2_ 188.9723, found: 188.9726.**1-Hydroxy-3-methylbutane-1,1-(H-phosphinylphosphonate) disodium salt 6b.** White powder (1.31 g, 95% yield (from pathway (e));); IR (neat, cm^−1^) ν = 3220br, 2956 w, 2315 w, 1185 s, 1110 m, 1048 s, 745 s; ^31^P{^1^H} NMR (162 MHz, D_2_O) δ 27.8 (d, *^2^J_P-P_* = 25.9 Hz), 16.6 (d, *^2^J_P-P_* = 25.9 Hz); ^31^P NMR (162 MHz, D_2_O) δ 27.8 (bd, *^1^J_P-H_* = 531.8 Hz, *^2^J_P-P_* = 25.9 Hz), 16.9–16.3 (m); ^1^H NMR (400 MHz, D_2_O) δ = 6.94 (dm, *^1^J_P-H_* = 531.8 Hz, 1H), 2.20–2.05 (m, 1H), 1.82–1.67 (m, 2H), 0.95 (d, *^3^J* = 6.6 Hz, 6H); ^13^C NMR (101 MHz, D_2_O) δ 75.9 (dd, *^1^J_P-C_* = 136.2 Hz, 92.1 Hz), 40.0, 24.4 (t, *^3^J_P-C_* = 8.0 Hz), 24.3, 24.2; MS (ESI^−^) *m*/*z*: 231.02 [M-H]^−^, 252.96 [M-2H+Na]^−^, 212.97 [M-H-H_2_O]^−^, 165.00 [M-H-H_3_PO_2_]^−^; HRMS (ESI^−^) *m*/*z*: [M-H]^−^ Calcd. for C_5_H_13_O_6_P_2_ 231.0193, found: 231.0195.**1-Hydroxyhexane-1,1-(H-phosphinylphosphonate) disodium salt 6c.** White powder (1.42 g, 98% yield (from pathway (e)); IR (neat, cm^−1^) ν = 3350 br, 2960 w, 2300 w, 1185 s, 1105 m, 961 s, 747 s; ^31^P{^1^H} NMR (162 MHz, D_2_O) δ 28.7 (d, *^2^J_P-P_* = 27.0 Hz), 15.2 (d, *^2^J_P-P_* = 27.1 Hz); ^31^P NMR (162 MHz, D_2_O) δ 28.7 (dm, *^1^J_P-H_* = 527.1 Hz, *^2^J_P-P_* = 27.0 Hz), 15.4–15.2 (m); ^1^H NMR (400 MHz, D_2_O) δ 7.01 (bd, *^1^J_P-H_* = 527.2 Hz, 1H, P-H), 1.94–1.65 (m, 2H), 1.59–1.44 (m, 2H), 1.35–1.19 (m, 4H), 0.85 (t, *^3^J* = 6.9 Hz, 3H); ^13^C NMR (101 MHz, D_2_O) δ 75.7 (dd, *^1^J_P-C_* = 130.9 Hz, 93.7 Hz), 32.6, 32.4, 23.4 (t, *^3^J_P-C_* = 6.5 Hz), 22.1, 13.5; MS (ESI^−^) *m*/*z*: 245.04 [M-H]^−^, 267.02 [M-2H+Na]^−^, 227.02 [M-H-H_2_O]^−^, 179.05 [M-H-H_3_PO_2_]^−^;HRMS (ESI^−^) *m*/*z*: [M-H]^−^ Calcd. for C_6_H_15_O_6_P_2_ 245.0349, found: 245.0351.**1-Hydroxy-1-phenylmethane-1,1-(H-phosphinylphosphonate) disodium salt 6d.** White powder (1.41 g, 95% yield (from pathway (b) with evaporation), 1.41 g, 95% yield (from pathway (e)); IR (neat, cm^−1^) ν = 3265 br, 2972 w, 2318 w, 1600 w, 1490 w, 1445 w, 1395 w, 1195 s, 1120 m, 975 m, 730 m; ^31^P{^1^H} NMR (162 MHz, D_2_O) δ 25.5 (d, *^2^J_P-P_* = 24.4 Hz), 14.2 (d, *^2^J_P-P_* = 24.4 Hz); ^31^P NMR (162 MHz, D_2_O) δ 25.5 (dd, *^1^J_P-H_* = 541.3 Hz, *^2^J_P-P_* = 24.4 Hz), 14.3 (d, *^2^J_P-P_* = 24.4 Hz); ^1^H NMR (400 MHz, D_2_O) δ 7.64 (d, *^3^J* = 7.6 Hz, 2H), 7.37 (t, *^3^J* = 7.5 Hz, 2H), 7.32–7.26 (m, 1H), 7.03 (d, *^1^J_P-H_* = 541.4 Hz, 1H, P-H); ^13^C NMR (101 MHz, D_2_O) δ 137.6, 127.9 (2C), 126.6 (2C), 125.8 (t, *^3^J_P-C_* = 4.1 Hz), 77.4 (dd, *^1^J_P-C_* = 135.9 Hz, 89.7 Hz); MS (ESI^−^) *m*/*z*: 250.99 [M-H]^−^, 272.97 [M-2H+Na]^−^, 232.98 [M-H-H_2_O]^−^, 185.00 [M-H-H_3_PO_2_]^−^; HRMS (ESI^−^) *m*/*z*: [M-H]^−^ Calcd. for C_7_H_9_O_6_P_2_ 250.9880, found: 250.9882.**1-Hydroxy-1-(4-tolyl)methane-1,1-(H-phosphinylphosphonate) disodium salt 6e.** White powder (1.43 g, 92% yield (from pathway (e)); IR (neat, cm^−1^) ν = 3235 br, 2975 w, 2900 w, 2310 w, 1649 w, 1510 w, 1336 w, 1188 s, 1095 s, 980 m, 754 m; ^31^P{^1^H} NMR (162 MHz, D_2_O) δ 26.0 (d, *^2^J_P-P_* = 24.8 Hz), 14.2 (d, *^2^J_P-P_* = 24.8 Hz); ^31^P NMR (162 MHz, D_2_O) δ 26.0 (dd, *^1^J_P-H_* = 538.8 Hz, *^2^J_P-P_* = 24.8 Hz), 14.2 (d, *^2^J_P-P_*= 24.8 Hz); ^1^H NMR (400 MHz, D_2_O) δ 7.56 (d, *^3^J* = 8.0 Hz, 2H), 7.24 (d, *^3^J* = 8.0 Hz, 2H), 7.05 (d, *^1^J_P-H_* = 539.9 Hz, 1H, P-H), 2.34 (s, 3H); ^13^C NMR (101 MHz, D_2_O) δ 136.4, 134.9, 128.3 (2C), 125.8 (2C), 77.6 (dd, *^1^J_P-C_* = 134.4 Hz, 91.0 Hz), 20.0; MS (ESI^−^) *m*/*z*: 265.00 [M-H]^−^, 286.99 [M-2H+Na]^−^, 246.99 [M-H-H_2_O]^−^199.02 [M-H-H_3_PO_2_]^−^; HRMS (ESI^−^) *m*/*z*: [M-H]^−^ Calcd. for C_8_H_11_O_6_P_2_ 265.0036, found: 265.0040.**1-Hydroxy-1-(4-methoxyphenyl)methane-1,1-(H-phosphinylphosphonate) disodium salt 6f.** White powder (1.34 g, 82% yield (from pathway (e)); IR (neat, cm^−1^) ν = 3250 br, 3000 w, 29020 w, 2315 w, 1611 w, 1511 m, 1464 w, 1192 s, 1118 m, 1034 m, 975 w, 740 w; ^31^P{^1^H} NMR (162 MHz, D_2_O) δ 25.7 (d, *^2^J_P-P_* = 26.4 Hz), 14.4 (d, *^2^J_P-P_* = 26.4 Hz); ^31^P NMR (162 MHz, D_2_O) δ 25.7 (dd, *^1^J_P-H_* = 539.2 Hz, *^2^J_P-P_* = 26.4 Hz), 14.4 (d, *^2^J_P-P_* = 26.4 Hz); ^1^H NMR (400 MHz, D_2_O) δ 7.59 (d, *^3^J* = 7.4 Hz, 2H), 7.01 (d, *^1^J_P-H_* = 539.2 Hz, 1H, P-H), 6.99 (d, *^3^J* = 7.8 Hz, 2H), 3.82 (s, 3H); ^13^C NMR (101 MHz, D_2_O) δ 157.5, 130.2, 127.2 (t, *^3^J_P-C_* = 4.2 Hz, 2C), 113.4 (2C), 77.1 (dd, *^1^J_P-C_* = 136.3 Hz, 90.3 Hz), 55.3; MS (ESI^−^) *m*/*z*: 281.00 [M-H]^−^, 302.98 [M-2H+Na]^−^, 262.99 [M-H-H_2_O]^−^, 215.01 [M-H-H_3_PO_2_]^−^; HRMS (ESI^−^) *m*/*z*: [M-H]^−^ Calcd. for C_8_H_11_O_7_P_2_ 280.9986, found: 280.9989.**1-Hydroxy-1-(4-fluorophenyl)methane-1,1-(H-phosphinylphosphonate) disodium salt 6g.** White powder (1.43 g, 91% yield (from pathway (e)); IR (neat, cm^−1^) ν = 3224 br, 2988 m, 2308 w, 1605 w, 1506 m, 1195 s, 1092 s, 1054 s, 973 w, 749 m; ^31^P{^1^H} NMR (162 MHz, D_2_O) δ 25.4 (d, *^2^J_P-P_* = 25.4 Hz), 14.0 (d, *^2^J_P-P_* = 24.7 Hz); ^31^P NMR (162 MHz, D_2_O) δ 25.4 (dd, *^1^J_P-H_* = 540.5 Hz, *^2^J_P-P_* = 26.4 Hz), 14.0 (d, *^2^J_P-P_* = 24.7 Hz); ^19^F{^1^H} NMR (377 MHz, D_2_O) δ −116.7–−116.8 (m); ^19^F NMR (377 MHz, D_2_O) δ −116.6–−116.8 (m); ^1^H NMR (400 MHz, D_2_O) δ 7.60–7.48 (m, 2H), 7.02 (t, *^3^J* = 8.6 Hz 2H), 6.94 (d, *^1^J_P-H_*
_=_ 540.6 Hz, 1H, P-H); ^13^C NMR (101 MHz, D_2_O) δ 161.6 (d, *^1^J_C-F_* = 242.8 Hz), 133.6, 127.8–127.4 (m, 2C), 114.4 (d, *^2^J_C-F_* = 21.2 Hz, 2C), 77.0 (dd, *^1^J_P-C_* = 135.2 Hz, 90.1 Hz); MS (ESI^−^) *m*/*z*: 268.98 [M-H]^−^, 290.96 [M-2H+Na]^−^, 250.97 [M-H-H_2_O]^−^, 202.99 [M-H-H_3_PO_2_]^−^; HRMS (ESI^−^) *m*/*z*: [M-H]^−^ Calcd. for C_7_H_8_FO_6_P_2_ 268.9786, found: 268.9787.**1-Hydroxy-1-(4-trifluoromethoxyphenyl)methane-1,1-(H-phosphinylphosphonate) disodium salt 6h.** White powder (1.62 g, 85% yield (from pathway (e)); IR (neat, cm-1) ν = 3320 br, 2319 w, 1650 w, 1610 w, 1192 s, 1114 s, 1070 s, 978 w, 761 m; ^31^P{^1^H} NMR (162 MHz, D_2_O) δ 25.2 (d, *^2^J_P-P_* = 23.1 Hz), 13.5 (d, *^2^J_P-P_* = 23.1 Hz); ^31^P NMR (162 MHz, D_2_O) δ 25.2 (dd, *^1^J_P-H_* = 541.9 Hz, *^2^J_P-P_* = 23.1 Hz), 13.5 (d, *^2^J_P-P_* = 23.1 Hz); ^19^F {^1^H} NMR (377 MHz, D_2_O) δ −57.7 (s); ^19^F NMR (377 MHz, D_2_O) δ −57.7 (bs); ^1^H NMR (400 MHz, D_2_O) δ 7.73–7.67 (m, 2H), 7.27 (d, *^3^J*
_=_ 8.5 Hz, 2H), 7.03 (bd, *^1^J_P-H_* = 541.9 Hz, 1H, P-H); ^13^C NMR (101 MHz, D_2_O) δ 147.5, 137.2, 127.3 (2C), 120.2 (2C), 120.3 (q, *^1^J_C-F_* = 256.8 Hz), 77.5 (dd, *^1^J_P-C_* = 133.5 Hz, 88.7 Hz); MS (ESI^−^) *m*/*z*: 334.97 [M-H]^−^, 356.95 [M-2H+Na]^−^, 316.96 [M-H-H_2_O]^−^, 268.98 [M-H-H_3_PO_2_]^−^; HRMS (ESI^−^) *m*/*z*: [M-H]^−^ Calcd. for C_8_H_8_F_3_O_7_P_2_ 334.9703, found: 334.9705.**1-Hydroxy-1-(2-tolyl)methane-1,1-bis(*H*-phosphonate) disodium salt 6i****.** The crude product was purified by simple washes (ethanol and methanol) to give a white powder (1.17 g, 75% yield (from pathway (e)); IR (neat, cm^−1^) ν = 3216 br, 2897 m, 2902 m, 2326 w, 1451 w, 1392 w, 1177 m, 1081 s, 976 w, 752 w; ^31^P{^1^H} NMR (162 MHz, D_2_O) δ 26.9–25.5 (m), 14.9 (d, *^2^J_P-P_* = 26.9 Hz); ^31^P NMR (162 MHz, D_2_O) δ 26.2 (dm, *^1^J_P-H_* = 547.8 Hz, *^2^J_P-P_* = 26.9 Hz), 14.9 (d, *^2^J_P-P_* = 26.9 Hz); ^1^H NMR (400 MHz, D_2_O) δ 7.64 (s, 1H), 7.10–7.02 (m, 3H), 7.06 (bd, *^1^J_P-H_* = 547.8 Hz, 1H, P-H), 2.46 (s, 3H, H_8_); ^13^C NMR (101 MHz, D_2_O) δ 137.7, 136.0, 132.6, 127.7, 126.7, 124.9, 80.2 (dd, *^1^J_P-C_* = 133.3 Hz, 88.0 Hz), 22.3; MS (ESI^−^) *m*/*z*: 265.00 [M-H]^−^, 286.99 [M-2H+Na]^−^, 246.99 [M-H-H_2_O]^−^, 199.02 [M-H-H_3_PO_2_]^−^; HRMS (ESI^−^) *m*/*z*: [M-H]^−^ Calcd. for C_8_H_11_O_6_P_2_ 265.0036, found: 265.0036.**1-Hydroxy-1-(2-methoxyphenyl)methane-1,1-(H-phosphinylphosphonate) disodium salt 6j.** White powder (1.06 g, 65% yield (from pathway (e)); IR (neat, cm^−1^) ν = 3675 w, 2989 w, 2901 w, 2310 w, 1581 w, 1488 w, 1465 w, 1205 s, 1118 m, 1058 w, 968 w, 758 w; ^31^P{^1^H} NMR (162 MHz, D_2_O) δ 24.7 (d, *^2^J_P-P_* = 28.7 Hz), 14.8 (d, *^2^J_P-P_* = 28.7 Hz); ^31^P NMR (162 MHz, D_2_O) δ 24.7 (dd, *^1^J_P-H_* = 561.5 Hz, *^2^J_P-P_* = 28.7 Hz), 14.8 (d, *^2^J_P-P_* = 28.7 Hz); ^1^H NMR (400 MHz, D_2_O) δ 7.57 (d, *^3^J* = 7.6 Hz, 1H), 7.22 (t, *^3^J* = 7.7 Hz, 1H), 7.14 (d, *^1^J_P-H_* = 561.5 Hz, 1H, P-H), 7.00–6.88 (m, 2H), 3.75 (s, 3H); ^13^C NMR (101 MHz, D_2_O) δ 157.0 (t, *^3^J_P-C_* = 4.4 Hz, 1C), 128.4, 127.7 (t, *^3^J_P-C_* = 4.6 Hz, 1C), 125.9, 120.7, 112.1, 78.3 (dd, *^1^J_P-C_* = 137.9 Hz, 89.6 Hz), 55.5; MS (ESI^−^) *m*/*z*: 281.00 [M-H]^−^, 302.98 [M-2H+Na]^−^, 262.99 [M-H-H_2_O]^−^, 215.01 [M-H-H_3_PO_2_]^−^; HRMS (ESI^−^) *m*/*z*: [M-H]^−^ Calcd. for C_8_H_11_O_7_P_2_ 280.9986, found: 280.9988.**1-Hydroxy-1-(1-napthyl)methane-1,1-(H-phosphinylphosphonate) disodium salt 6l.** White powder (1.83 g, 53% yield (from pathway (e)); IR (neat, cm^−1^) ν = 3312 br, 2372 w, 2995 w, 1181 s, 1095 m, 967 m, 785 s; ^31^P{^1^H} NMR (162 MHz, D_2_O) δ 28.1–23.2 (m), 14.6 (d, *^2^J_P-P_* = 25.6 Hz); ^31^P NMR (162 MHz, D_2_O) δ 25.7 (dm, *^1^J_P-H_* = 544.5 Hz), 14.6 (d, *^2^J_P-P_* = 25.6 Hz); ^1^H NMR (400 MHz, D_2_O) δ 8.92–8.83 (m, 1H), 7.87 (s, 1H), 7.80–7.73 (m, 1H), 7.69 (d, *^3^J* = 7.9 Hz, 1H), 7.43–7.30 (m, 3H), 7.24 (d, *^1^J_P-H_* = 545.1 Hz, 1H, P-H); ^13^C NMR (101 MHz, D_2_O) δ 134.1, 132.1–131.3 (m, 2C), 128.4, 128.3 (2C), 127.8–127.4 (m), 125.5–125.1 (m, 2C), 124.8, 80.6 (dd, *^1^J_P-C_* = 132.1 Hz, 86.9 Hz); MS (ESI^−^) *m*/*z*: 301.00 [M-H]^−^, 322.99 [M-2H+Na]^−^, 282.99 [M-H-H_2_O]^−^, 235.02 [M-H-H_3_PO_2_]^−^; HRMS (ESI^−^) *m*/*z*: [M-H]^−^ Calcd. for C_11_H_11_O_6_P_2_ 301.0036, found: 301.0038.**1-Hydroxy-1-(2-thienyl)methane-1,1-(H-phosphinylphosphonate) disodium salt 6m.** White powder (1.24 g, 82% yield (from pathway (e)); IR (neat, cm^−1^) ν = 3305 br, 2961 w, 2308 w, 1666 w, 1423 w, 1434 w, 1369 w, 1190 s, 1095 m, 972 w, 725 w; ^31^P{^1^H} NMR (162 MHz, D_2_O) δ 24.6 (d, *^2^J_P-P_* = 24.0 Hz), 12.7 (d, *^2^J_P-P_* = 24.0 Hz); ^31^P NMR (162 MHz, D_2_O) δ 24.6 (dd, *^1^J_P-H_* = 542.5 Hz, *^2^J_P-P_* = 24.0 Hz), 12.7 (d, *^2^J_P-P_* = 24.0 Hz); ^1^H NMR (400 MHz, D_2_O) δ 7.37–7.30 (m, 1H), 7.21–7.11 (m, 1H), 7.08–7.03 (m, 1H), 7.01 (bd, *^1^J_P-H_* = 542.5 Hz, 1H, P-H); ^13^C NMR (101 MHz, D_2_O) δ 142.8, 127.0 (t, *^4^J_P-C_* = 2.6 Hz), 123.8 (t, *^5^J_P-C_* = 2.5 Hz), 123.7 (t, *^3^J_P-C_* = 5.4 Hz), 77.2 (dd, *^1^J_P-C_* = 133.1 Hz, 90.6 Hz); MS (ESI^−^) *m*/*z*: 256.94 [M-H]^−^, 278.93 [M-2H+Na]^−^, 238.93 [M-H-H_2_O]^−^, 190.96 [M-H-H_3_PO_2_]^−^; HRMS (ESI^−^) *m*/*z*: [M-H]^−^ Calcd. for C_5_H_7_O_6_P_2_S 256.9444, found: 256.9447.**1-Hydroxy-2-phenylethane-1,1-(H-phosphinylphosphonate) disodium salt 6n.** White powder (1.40 g, 90% yield; IR (neat, cm^−1^) ν = 3289 br, 2989 m, 2901 m, 2315 w, 1187 s, 1121 m, 1056 m, 756 m; ^31^P{^1^H} NMR (162 MHz, D_2_O) δ 26.7 (d, *^2^J_P-P_* = 22.4 Hz), 15.8 (d, *^2^J_P-P_* = 21.9 Hz); ^31^P NMR (162 MHz, D_2_O) δ 27.8 (bd, *^1^J_P-H_* = 533.8 Hz, *^2^J_P-P_* = 20.3 Hz), 16.0–15.7 (m); ^1^H NMR (400 MHz, D_2_O) δ 7.48–7.40 (m, 2H), 7.36–7.18 (m, 3H), 6.88 (dm, *^1^J_P-H_* = 534.7 Hz, 2H, P-H), 3.38–3.07 (m, 2H); ^13^C NMR (101 MHz, D_2_O) δ 137.1–136.2 (m), 131.4 (2C), 127.9 (2C), 126.4, 74.5 (dd, *^1^J_P-C_* = 134.4 Hz, 94.0 Hz), 37.2; MS (ESI^−^) *m*/*z*: 265.00 [M-H]^−^, 286.99 [M-2H+Na]^−^, 246.99 [M-H-H_2_O]^−^; HRMS (ESI^−^) *m*/*z*: [M-H]^−^ Calcd. for C_8_H_11_O_6_P_2_ 265.0036, found: 265.0045.**1-Hydroxy-1-(3-azidopropyl)methane-1,1-(H-phosphinylphosphonate) disodium salt 6o**. White powder (1.26 g, 83% yield (from pathway (e)); IR (neat, cm^−1^) ν = 3305 br, 2964 w, 2310 w, 1665 w, 1421 w, 1432 w, 1369 w, 1192 s, 1097 m, 972 w, 724 w; ^31^P{^1^H} NMR (162 MHz, D_2_O) δ 27.4 (d, *^2^J_P-P_* = 27.3 Hz), 16.0 (d, *^2^J_P-P_* = 27.3 Hz); ^31^P NMR (162 MHz, D_2_O) δ 27.4 (dm, *^1^J_P-H_* = 532.4 Hz, *^2^J_P-P_* = 27.3 Hz), 16.7–15.8 (m); ^1^H NMR (400 MHz, D_2_O) δ 6.95 (bd, *^1^J_P-H_* = 532.4 Hz, 1H, P-H), 3.32 (t, *^3^J* = 6.0 Hz, 2H), 2.01–1.74 (m, 4H); ^13^C NMR (101 MHz, D_2_O) δ 74.4 (dd, *^1^J_P-C_* = 135.7 Hz, 92.4 Hz), 51.8, 29.3, 23.1 (t, *^3^J_P-C_* = 6.7 Hz); MS (ESI^−^) *m*/*z*: 258.01 [M-H]^−^, 279.99 [M-2H+Na]^−^, 239.99 [M-H-H_2_O]^−^, 214.99 [M-H-N_3_]^−^, 192.02 [M-H-H_3_PO_2_]^−^; HRMS (ESI^−^**)** *m*/*z*: [M-H]^−^ Calcd. for C_4_H_10_N_3_O_6_P_2_ 258.0050, found: 258.0053.**1-Hydroxy-1-(5-azidopentyl)methane-1,1-(H-phosphinylphosphonate) disodium salt 6p**. White powder (1.34 g, 85% yield (from pathway (e)); IR (neat, cm^−1^): ν = 3328 br, 2956 w, 2303 w, 2121 w, 1182 s, 1105 m, 959 m, 746 w. ^31^P{^1^H} NMR (162 MHz, D_2_O) δ 27.4 (d, *^2^J_P-P_* = 27.5 Hz), 16.0 (d, *^2^J_P-P_* = 27.5 Hz); ^31^P NMR (162 MHz, D_2_O) δ 27.4 (dm, *^1^J_P-H_* = 530.9 Hz, *^2^J_P-P_* = 27.5 Hz), 17.1–16.0 (m); ^1^H NMR (400 MHz, D_2_O) δ 6.95 (bd, *^1^J_P-H_* = 530.9 Hz, 1H, P-H), 3.30 (t, *^3^J* = 7.0 Hz, 2H), 1.92–1.75 (m, 2H), 1.67–1.48 (m, 4H), 1.41–1.28 (m, 2H); ^13^C NMR (101 MHz, D_2_O) δ 74.8 (dd, *^1^J_P-C_* = 135.8 Hz, 91.9 Hz), 51.2, 32.8, 27.8, 26.9, 22.9 (t, ^3^*J_P-C_* = 6.1 Hz); MS (ESI^−^) *m*/*z*: 286.04 [M-H]^−^, 308.02 [M-2H+Na]^−^, 268.03 [M-H-H_2_O]^−^, 243.0194 [M-H-N_3_]^−^, 220.05 [M-H-H_3_PO_2_]^−^; HRMS (ESI^−^) *m*/*z*: [M-H]^−^ Calcd. for C_6_H_14_N_3_O_6_P_2_ 286.0363, found: 286.0364.

#### 3.2.5. Synthesis of Alendrionate **14o** and Neridrionate **14p** by Hydrogenolysis

Pd/C (10%, 6.00 mg) was added to a solution of **6** (0.21 mmol, 1.00 equiv.) in H_2_O/MeOH: 1/1 (4.00 mL). The system was flushed with a dihydrogen balloon. The reaction mixture was stirred under a dihydrogen atmosphere for 2 h. The end of the reaction was monitored by ^31^P NMR spectroscopy. Then, the mixture was filtered over a pad of celite^®^. After solvent evaporations, the residue was dissolved in water (2.00 mL) and an aqueous solution of sodium hydroxide 1 M was added until pH = 7. The solution was lyophilized. The crude product was purified by simple washes (ethanol and methanol) to give a white powder.

**1-Hydroxy-1-(3-aminopropyl)methane-1,1-(H-phosphinylphosphonate) disodium salt 14o**.White powder (0.047 g, 87% yield); IR (neat, cm^−1^): ν = 3319 br, 2951 w, 2309 w, 1204 w, 1178 s, 1102 m, 975 m, 737 w; ^31^P{^1^H} NMR (162 MHz, D_2_O) δ 26.8 (d, *^2^J_P-P_* = 27.3 Hz), 15.8 (d, *^2^J_P-P_* = 27.3 Hz); ^31^P NMR (162 MHz, D_2_O) δ 26.8 (dm, *^1^J_P-H_* = 535.6 Hz), 16.7–16.5 (m); ^1^H NMR (400 MHz, D_2_O) δ 6.95 (bd, *^1^J_P-H_*= 535.6 Hz, 1H, P-H), 3.13–2.82 (m, 2H), 2.06–1.70 (m, 4H); ^13^C NMR (101 MHz, D_2_O) δ 73.9 (dd, *^1^J_P-C_* = 140.7 Hz, 94.4 Hz), 39.9, 28.7, 21.8 (m); MS (ESI^−^) *m*/*z*: 232.01 [M-2H]^−^; HRMS (ESI^−^) *m*/*z*: [M-2H]^−^ Calcd. for C_4_H_12_NO_6_P_2_ 232.0145, found: 232.0148.**1-Hydroxy-1-(5-aminopentyl)methane-1,1-(H-phosphinylphosphonate) disodium salt 14p.** White powder (0.050 g, 84% yield); IR (neat, cm^−1^): ν = 3320 br, 2951 w, 2309 w, 1205 w, 1179 s, 1101 m, 974 m, 738 w; ^31^P{^1^H} NMR (162 MHz, D_2_O) δ 27.0 (d, *^2^J_P-P_* = 28.6 Hz), 16.8 (d, *^2^J_P-P_* = 29.4 Hz); ^31^P NMR (162 MHz, D_2_O) δ 27.0 (dm, *^1^J_P-H_* = 532.4 Hz), 17.3–16.4 (m); ^1^H NMR (400 MHz, D_2_O) δ 7.00 (bd, *^1^J_P-H_* = 532.4 Hz, 1H, P-H), 3.05 (bs, 2H), 2.00–1.78 (m, 2H), 1.85–1.59 (m, 4H), 1.56–1.32 (m, 2H); ^13^C NMR (101 MHz, D_2_O) δ 74.4 (dd, *^1^J_P-C_* = 141.0 Hz, 88.1 Hz), 39.2, 31.7, 26.3, 26.2, 22.5 (t, ^3^*J_P-C_* = 6.2 Hz); MS (ESI^−^) *m*/*z*: 260.05 [M-H]^−^, 282.03 [M-2H+Na]^−^, 242.04 [M-H-H_2_O]^−^, 194.06 [M-H-H_3_PO_2_]^−^; HRMS (ESI^−^) *m*/*z*: [M-H]^−^ Calcd. for C_2_H_16_NO_6_P_2_ 265.0458, found: 260.0459.

### 3.3. Complexation to Hydroxyapatite (HA)

At 37 °C, 600 µL of a 5 mM solution of alendronate, alendrinate or alendrionate **14o** in PBS 1X (phosphate-buffered saline, pH = 7.4) and an excess of HA (12 mg; 8 eq.) were stirred for 3 h. At t = 15, 70 and 180 min, the mixture was centrifuged at 2500 rpm for 5 min. The supernatant was recovered and analyzed by ^31^P NMR without lock and shim settings. The binding rates were then determined by comparing the relative integration of free phosphorylated compounds with PO_4_^2−^ (δ = 0 ppm) used as internal reference.

### 3.4. Cytotoxicity Assay

MDA-MB-231 (human breast adenocarcinoma), MIA PaCa-2 (human pancreatic carcinoma) and A549 (human lung carcinoma) cell lines were obtained from the American Type Culture Collection (ATCC). Cells were cultured in high glucose Dulbecco’s modified Eagle’s medium (DMEM) supplemented with 10% fetal bovine serum (FBS) and 1% penicillin/streptomycin at 37 °C in a 5% CO_2_ humidified atmosphere. DMEM culture medium, FBS and all reagents used in biological study were purchased from Sigma Aldrich.

### 3.5. In Vitro Antiproliferative Activity

Cancer cell growth inhibition was evaluated using the MTT microculture tetrazolium assay. Cells were seeded in a 96-well plate in the supplemented culture medium at a density of 4 × 10^3^ cells/well for MDA-MB-231 cells and 1 × 10^3^ cells/well for MIA PaCa-2 and A549 cell lines. After 24 h incubation, the culture medium was removed and replaced with 100 µL of supplemented culture medium containing increasing concentrations of evaluated freshly synthesized compound (0, 1, 5, 10, 50, 100, 200, 500 µM, 1 and 5 mM). Cells were incubated at 37 °C for 72 h. Then, 10 µL of a MTT solution (5 mg/mL in PBS) were added to each well. After 3 more hours of incubation at 37 °C, the culture medium was removed and 50 µL/well of DMSO were introduced to dissolve the insoluble purple formed product. Optical density was measured at 570 nm using a Thermo Scientific Multiskan FC microplate reader. IC_50_ values (concentration of the compound where the response is reduced by half) were determined by using GraphPad software. All experiments were carried out 3 times in triplicate.

## 4. Conclusions

We have developed an efficient methodology that allows for the preparation of original hydroxymethylene(phosphinyl)phosphonates **6**. Our research is a considerable advance in the synthesis of hydroxymethyl(phosphinyl)phosphonate scaffold, which is very poorly described in the literature. Moreover, the optimization of our procedure allows us to highlight a one-pot synthesis in which no purification of intermediate species is necessary. Various HMPPs **6** and **14** were obtained starting from functionalized or not aliphatic or substituted (hetero)aromatic substrates **2**. Good to excellent yields (53–98%) were observed after easy purification and short reaction times (20 min–2.5 h for each reagent addition). Biologically relevant aminoalkyl compounds alendrionate **14o** and neridrionate **14p** were also synthesized. Their in vitro antiproliferative efficiency on cancer cell lines was evaluated and compared to bisphosphonate and bisphosphinate analogues. They showed interesting and encouraging results, especially neridrionate **14p** on A549 cells (IC_50_ = 26.6 µM), since the change in pharmacophore allows to maintain a biological activity. These preliminary results will prompt us to synthesize third generation HMBP analogues which would give more potent inhibition. In addition, the weaker and slower complexation of HMPP alendrionate **14o** to hydroxyapatite offers good hope for future via studies aimed at targeting soft tissue tumors rather than bone.

## Data Availability

The data presented in this study are available in article or Appendix A.

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
