# Peer review of "One-Pot Synthesis of Phosphinylphosphonate Derivatives and Their Anti-Tumor Evaluations"

_molecules, 2021, doi:10.3390/molecules26247609_

Round 1

Reviewer 1 Report

The manuscript reporting on the elaboration, 31P-NMR-supported monitoring and optimization of a one-pot synthesis of biologically relevant phosphinylphosphonates. The formation of isomer products types 6 and 9 is correctly interpreted in terms of a plausible reaction mechanism. Improving the general quality of the work, addressing potentiantial treatment of bone pathologies and different tumors, the Authors performed a Ca-complexation study and an MTT-based  antiproliferative evaluation of selected products on three agressive human malignant cell lines. Although this manuscript is worth to be published in Molecules, prior to final acceptance the following points need to be addressed in the revised version. Being too crowded, Schemes 1 and 2 should be reformulated using more chemical structures with numbering of the compounds. Accordingly, in the text below Scheme 1 the compounds should be presented with their numbers. Intermediate 8a contains 2 stereogenic centers pointing to the possibility of its existance as two diastereomers. However, "Spectrum e" suggests the presence of one diastereomer adopting a well-defined conformation stabilised by intramolecular stereoelectronic Si--O interactions. The authors are expected to briefly comment on the diastereoselective formation of 8a

Author Response

Comments of reviewer 1

The manuscript reporting on the elaboration, 31P-NMR-supported monitoring and optimization of a one-pot synthesis of biologically relevant phosphinylphosphonates. The formation of isomer products types 6 and 9 is correctly interpreted in terms of a plausible reaction mechanism. Improving the general quality of the work, addressing potentiantial treatment of bone pathologies and different tumors, the Authors performed a Ca-complexation study and an MTT-based  antiproliferative evaluation of selected products on three agressive human malignant cell lines. Although this manuscript is worth to be published in Molecules, prior to final acceptance the following points need to be addressed in the revised version.

Being too crowded, Schemes 1 and 2 should be reformulated using more chemical structures with numbering of the compounds. Accordingly, in the text below Scheme 1 the compounds should be presented with their numbers.

Answer: In Figure 1 and Scheme 1, compounds have been numbered and molecule numbers have been introduced in the text of part 1 “Introduction”. Compounds were already numbered in Scheme 2. The reviewer 1 certainly wanted to mention Figure 1 and Scheme 1 instead of Schemes 1 and 2. Moreover, Figure 1 and Schemes 1 and 2 have been reorganized for more clearness.

Intermediate 8a contains 2 stereogenic centers pointing to the possibility of its existance as two diastereomers. However, "Spectrum e" suggests the presence of one diastereomer adopting a well-defined conformation stabilised by intramolecular stereoelectronic Si--O interactions. The authors are expected to briefly comment on the diastereoselective formation of 8a

Answer: As well explained by the reviewer 1, silylated HMPP 8a presents 2 stereogenic atoms and could exit as 2 diastereoisomers in NMR spectrum. Two peaks about 20 and 24 ppm corresponding to the stereogenic phosphorus could indicate the presence of the 2 diastereoisomers. In the other hand, the signals are also very large probably due to silyl group equilibrium. So, in our opinion, it is difficult to conclude about the presence of one or two diastereoisomers.

Reviewer 2 Report

The reviewed paper concerns the synthesis and cytotoxic activity of phosphinylphosphonate derivatives. It is a very well-written publication that extends the current knowledge. Well-thought-out and precisely described experiments (both in the discussion and in the experimental part) deserve attention. The work is also nicely prepared in terms of graphics. These are the main advantages of this article. However, many of the compounds obtained seem to be quite heavily contaminated - which is especially visible in the 31P NMR spectra (see SI, 6e, 6h, 6i, 6l-p, 14o, and 14p). How can you explain this? Is there any chance to obtain products of higher purity (maybe recrystallization?) and then better spectra?

I have a few more questions and comments listed below:

Questions:

Have you tested the cytotoxicity of the obtained compounds 6? Would it make sense based on your experience?

Comments:

Keywords: - I suggest „phosphinylphosphonate” instead of „phosphinylphosphinate”

Line 169 – Consider R= n-pentyl (amyl) instead of R = pent

Line 375 and later in the text – consider removing „an addition funnel”

Line 408 (430, 443) – consider „quenched with methanol and the mixture was stirred for 30 minutes” not „quenched with methanol for 30 minutes”

Spectroscopic characterization:

- please check if the P-H signal (1H NMR, 7.1-6.9 ppm) really is dm and not d (compounds 6b, 6c, 6h, 6i, 6m, 6n, 6o, 6p, 14o, and 14p)

- compound 6b: 13C NMR – 24,3 consider two singlets (distereotopic carbons) instead of doublet

- how about IR spectra?

SI

General: some spectra need to be phased (I mean phase correction: S5, S11( 31P NMR), S16, S22, S38, S41, S53)

I can recommend this work for publication after minor corrections and clarification of doubts related to the purity of compounds.
